# Intra-horn insemination in the alpaca *Vicugna pacos*: Copulatory wounding and deep sperm deposition

**Patricia L. R. Brennan**[1]*, **Stephen Purdy**[2], **Sarah J. Bacon**[1]

1 Department of Biological Sciences, Mount Holyoke College, South Hadley, MA, United States of America,
2 North American Camelid Studies Program, Nunoa Project, Belchertown, MA, United States of America

* pbrennan@mtholyoke.edu

**Data Availability Statement:** All relevant data are within the manuscript and its Supporting Information Files.

**Funding:** Funding was provided by MHC as a faculty grant to PLRB. The Brennan Lab is funded

## Abstract

Alpacas (*Vicugna pacos*) are reported to be the rare mammal in which the penis enters the uterus in mating. To date, however, only circumstantial evidence supports this assertion. Using female alpacas culled for meat, we determined that the alpaca penis penetrates to the very tips of the uterine horns, abrading the tract and breaking fine blood vessels. All female alpacas sacrificed one hour or 24 hours after mating showed conspicuous bleeding in the epithelium of some region of their reproductive tract, including the hymen, cervix and the tips of each uterine horn, but typically not in the vagina. Unmated females showed no evidence of conspicuous bleeding. Histological examination of mated females revealed widespread abrasion of the cervical and endometrial epithelium, injuries absent in unmated females. Within one hour of mating, sperm were already present in the oviduct. The male alpaca's cartilaginous penis tip with a hardened urethral process is likely responsible for the copulatory abrasion. The entire female reproductive tract interacts with the penis, functioning like a vagina. Alpacas are induced ovulators, and wounding may hasten delivery of the seminal ovulation-inducing factor beta-NGF into the female's blood stream. There is no evidence of sexual conflict in copulation in alpaca, and thus wounding may also be one of a variety of mechanisms devised by mammals to induce a beneficial, short-term inflammatory response that stimulates blastocyst implantation, the uterine remodeling associated with placental development, and thus the success of early pregnancy.

## Introduction

The depth of penile intromission during copulation in mammals has important consequences to the evolution of female immune defenses in the reproductive tract, the coevolution of genitalia in males and females, the evolution of ejaculate traits that can help to propel the sperm to the sites of storage and fertilization, and the preparation of the uterus for implantation. Yet these questions have seldom been examined from these perspectives, because in most mammals, penile intromission is limited to the vagina, with insemination occurring primarily in the cranial vagina or fornix [1]. In two well-known exceptions, the ejaculate is propelled

by NSF CAREER grant IOS: 2042260. The funders had no role in study design, data collection and analysis, decision to publish, or preparation of the manuscript.

**Competing interests:** The authors have declared that no competing interests exist.

through the cervix, but penile penetration is limited to the vagina. In the horse (*Equus equus*), the glans penis presses against the cervix, and the ejaculate is pushed through the cervix to the entrance of the uterus and therefore horses are classified as intrauterine inseminators. In the dog (*Canis domesticus*), copulation involves a lock resulting from inflation of the *bulbus glandis* in the penile shaft that forces the ejaculate through the cervix into the uterus [2]. A third variant is intracervical penile intromission with intrauterine insemination, reported in pigs (*Sus scrofa domesticus*), in which the spiral glans penis locks in the folds of the cervix and males ejaculate in the uterus [3, 4]. Other species including mice (*Mus musculus*), rats (*Rattus norvegicus*) and ferrets (*Mustela putorius*) have been classified as intrauterine inseminators [5], but no details of how this uterine insemination occurs are found in the literature.

Alpacas (*Vicugna pacos*) are unique in that reportedly they have intrauterine penile intromission with intrauterine insemination [6]. This mode of insemination would be unusual indeed because it requires that the penis pass through the whole cervix to deposit sperm directly in the uterus. One important function of the cervix (particularly the cervical mucus) is to protect the uterus and upper female reproductive tract from ascending infection [7]. The cervix is typically in a closed state, and abundant mucus production catches any potential pathogens that may enter with the penis or ejaculate [8]. Intromission that compromises this important barrier function would seem to put the female alpaca at increased risk of ascending infection of the reproductive tract.

The ejaculate of alpacas is also unusual; alpacas are considered to be "dribble" ejaculators, meaning males ejaculate small quantities of sperm during copulation tied with urethral pulses [9]. The average breeding time for alpacas is 20 minutes, yet less than 2 mL of sperm are often reported during semen collection. Semen collection every five minutes during copulation showed that the concentration of sperm is 63–176 million sperm/mL, with sperm concentration increasing significantly towards the end of copulation [10]. For comparison, a stallion will produce 50–100 mL of sperm and ejaculate several billion sperm per copulation [3].

While the alpaca literature often repeats intrauterine intromission as factual, the main source of this observation is an abstract from a camelid conference in Peru [11] that is unavailable from any library or personal contact these authors could obtain. The only other confirmatory source is Bravo et al [6] who described the uterus after copulation in alpaca as "inflamed, edematous and hyperemic", and assumed the source to be the penis moving inside the uterus. To find evidence for or against intrauterine intromission and ejaculation in alpaca, we first described the genitalia of male and female alpacas to determine if it was even possible for the penis to reach all the way to the uterus by going through the cervix. We found evidence that while the vagina is very long, the penis is very long and thin and likely capable of reaching the uterus [12]. Here we report on a series of opportunistic mating experiments designed to determine the depth of penile penetration and the likely site of insemination. We examined the lumen of the reproductive tract in receptive but unmated, and mated female alpacas (euthanized 1hr and 24 hrs postmating) to identify the location of sperm immediately after mating and evaluate evidence of wounding and bleeding within the female tract.

## Materials and methods

We obtained whole reproductive tracts from adult female alpacas that were being culled for meat processing at a farm in Connecticut, USA, where animals are maintained in accordance with State and federal guidelines concerning livestock management for small farms. The number of females in each treatment was limited by the farmer's ability to process the carcasses we required each time we visited the farm, depending on his schedule and availability of adult females suitable for this study. Females' reproductive receptivity was assessed by exposing

them to an intact male. If the female was receptive she would adopt the prone position. For (a) receptive but unmated females, the farmer euthanized the females and removed their reproductive organs including the ovaries, uterine horns, uterus, cervix, vagina, and external genitalia (labia and commissure), and we examined them immediately after euthanasia. For mated females we asked the farmer to mate females either (b) immediately before euthanasia or (c) 24 hours before euthanasia. Females mated immediately before euthanasia were maintained in the down breeding position after death. The farmer made a lateral incision in their abdomen as a point of entry to place ties between the uterine horns and the uterus, between the uterus and the cervix, and between the cervix and the vagina, to minimize any ejaculate movement prior to examination. The farmer removed the reproductive organs and we examined them and photograph them immediately after removal. For the females mated 24 hrs before euthanasia, he removed the whole female tract for examination during the usual processing for meat.

Upon removal from the body, the reproductive tract was cleaned of excess connective tissue, straightened and photographed. We opened the lumen of the tract from the dorsal aspect and examined the tissue and photographed all the regions of the reproductive tract. We then used a glass slide to touch each region of the inside of the reproductive tract and stained these slides either with a modified Giemsa stain (Differential Quik Stain Kit, VWR, Radnor PA) to look for the presence of inflammatory cells (unmated and mated +24 hrs) and sperm (mated +24 hrs), or with Semen Morphology Stain (eosin/nigrosin) (mated +1hr). While this technique is not quantitative because of our inability to standardize the area of the tissue we were pressing on, or the pressure applied, it can provide information on the presence/absence and relative abundance of sperm and immune cells (neutrophils and lymphocytes) each of which was calculated on a scale of 0–4, where 1 = 1–25% of the slide was covered with sperm/immune cells, 2 = 25–50%, 3 = 50–75% and 4 = 75–100%.

All photographs taken of the reproductive tract gross anatomy were analyzed using ImageJ [13], to count the number of pixels with blood to quantify the extent of the abrasion to the female tract in six unmated females, four females mated within the hour (two had to be excluded because one was pregnant, and the other had an apparent uterine infection], and six females mated 24 hrs prior. We counted pixels with and without blood separately for each of the vagina, hymen, cervix, uterus, and left and right uterine horns. We then calculated the percent of pixels with hematoma vs the percent without and compared across all categories. Using R Studio 4.2.2 [14], we ran Welch's ANOVA on the percentage of bloody pixels data to account for unequal variance, followed by a Tukey Pairwise test comparing mated on the day of euthanasia (mated + 1H), mated the day prior (mated +24 H), and unmated to determine which comparisons were significantly different.

We dissected a small piece of tissue from each region of the reproductive tract, including mucosa and submucosa of vagina, cervix, uterine body and horns, and oviductal papilla for histological examination (n = 4 receptive unmated females, n = 4 mated + 1hr, n = 6 mated +24 hrs). Samples for histology were fixed in neutral buffered formalin for 24 hours then transferred to 70% EtOH until paraffin infiltration, embedding, sectioning (4μm), and staining with both H&E and Masson's Trichrome [15]. We examined the slides and took photomicrographs (CellSens, Olympus) on an Olympus BX51 light microscope to examine mucosal abrasion.

Subepithelial neutrophil infiltration in these H&E-stained sections was measured by manually counting neutrophils associated with the endometrial epithelium over a linear distance of 1.5mm. Slides were assessed by a single observer blind to experimental condition, and at least 2 semi-serial sections were counted for each alpaca. Values were averaged across sections within each experimental condition. Two alpacas in the mated + 24 hrs condition had to be excluded due to tissue overfixation. Cells were coded as neutrophils based on their pinkish,

eosinophilic cytoplasm and lobed nucleus. Neutrophils either within the epithelium or up to two cell diameters below the epithelium's basal surface were counted. Representative neutrophil infiltration was photographed at 400x on an Olympus BX41 light microscope using Pixelink OEM (Navitar).

### Ethical statement

This study was approved by the Mount Holyoke College IACUC (Institutional Animal Care and Use Committee). Though the authors did not directly handle live animals for this research, all work was performed under IACUC permit #BR-59-0820 through Mount Holyoke College. Animals were euthanized by the farmer following State and federal guidelines for culling livestock for meat production in small farms.

## Results

### Blood visible grossly in the tract

Mated females had much higher percentages of bloody pixels than receptive unmated females (Fig 1A and 1B). While there was a lot of variation among females in the percent of bloody pixels present per region (Fig 2), the left and/or right uterine horns had the highest percentage of bloody pixels in 6 of 10 females, while the cervix and uterus had the highest percentages in 3 of 10 females (data in S1 Table). The vagina had the lowest evidence of blood in the reproductive tract in both mated categories but one mated female (F036), had blood only in the hymen and vagina and none in the cervix, uterus or uterine horns (Fig 2, all data in S1 Table, and photos in S1 Fig). The hemorrhagic tissue was still evident 24 hrs after mating. The Welch ANOVA

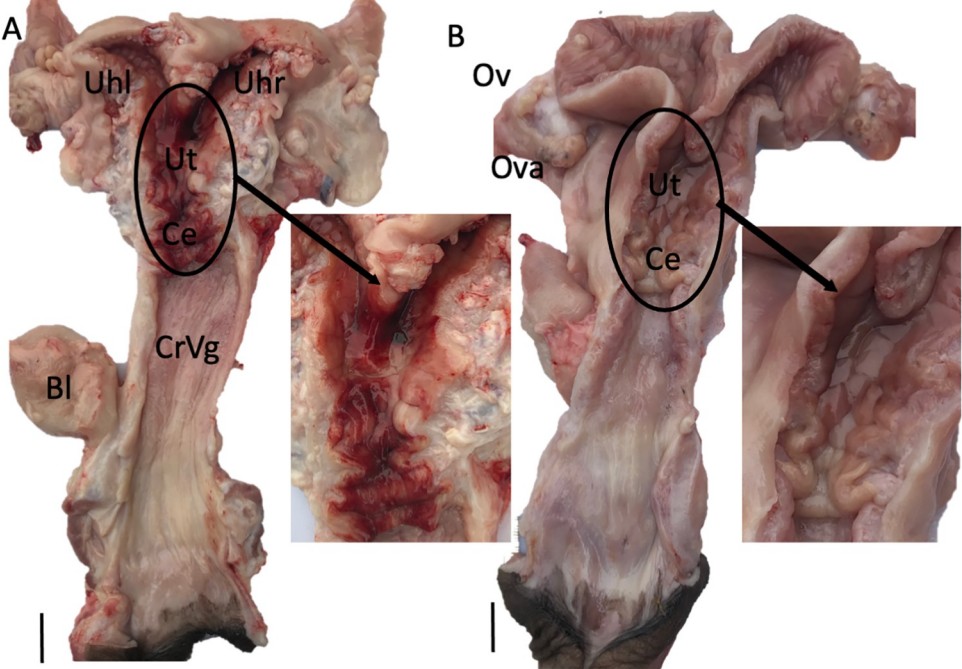

**Fig 1.** Dorsal view of the reproductive tract of a female alpaca: Mated + 1 h (A), and unmated alpaca (B). The insets are magnified views of the cervical and uterine region in both animals. The mated female has extensive hemorrhage primarily in the cervix, uterus, and uterine horns, while the vagina is largely untouched. The unmated female (B) shows no sign of bleeding. Bl: Bladder, CrVg: cranial vagina, Ce: Cervix, Ut: Uterus, Uhl: Uterine horn left, Uhr: Uterine horn right, Ov: Oviduct, Ova: Ovary. Scale Bar: 2 cm.

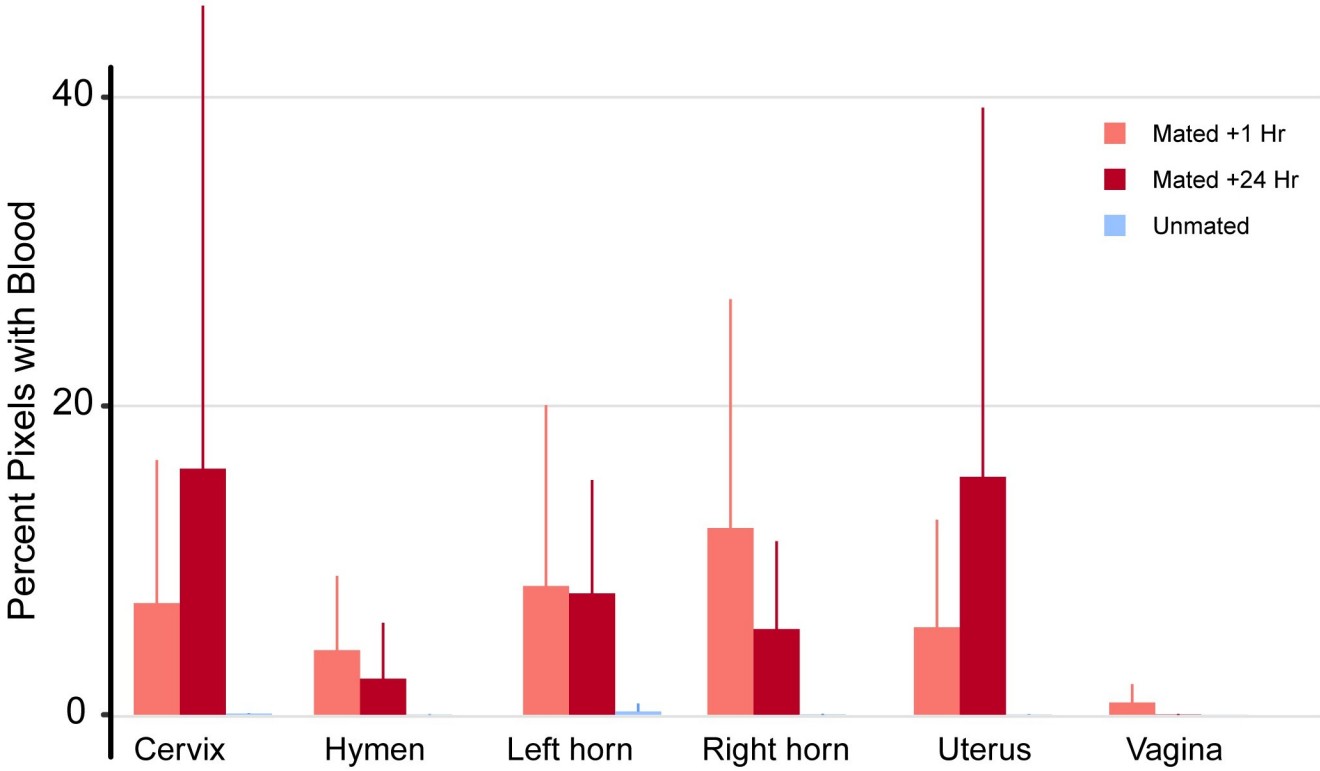

**Fig 2. Percentages of red pixels indicative of blood per region of the female reproductive tract in alpacas.** All females had minimal blood in the vagina, but females mated within an hour and 24 hrs prior to euthanasia had abundant blood in their upper reproductive tracts (Uterus, and uterine horns), while variability was high among females (Average +/- Std Error).

comparing bloody pixels in mated vs unmated females was significant (data: Percent_blood and Mating_status F = 10.64, num df = 2, denom df = 37.012, p-value = 0.0002). The Tukey pairwise test confirmed that while the percent of bloody pixels in females examined the day they mated, vs. females examined the day after they mated was no different, the comparison between unmated females with either mated females + 1 h vs. mated females + 24 h was significant, with mated females in both categories having significantly higher percentages of bloody pixels in their reproductive tract than unmated females (Fig 3).

## Tissue abrasion and neutrophil infiltration

In tissue samples taken from female alpacas within the first hour after mating, the epithelium lining the vagina remained intact, though in one sample we saw minor subepithelial suffusion of blood. The vaginal epithelium was intact 24 hours after mating, similar to its condition in unmated, reproductively receptive females (Fig 4). In contrast, the epithelium lining the cervix was torn and fragmented within an hour of mating, and subepithelial hemorrhage was visible (Fig 5). Twenty-four hours later the cervical epithelium was intact, similar to the unmated condition.

In the uterus, the epithelium was scuffed and patchy in the hour after mating with areas of hemorrhage visible in the underlying stromal tissue (Fig 5). This mating-associated wounding extended into both uterine horns (Fig 5). Twenty-four hours later, the epithelium was intact though some red blood cells remained trapped in the underlying tissue. At the tip of each uterine horn a small papilla (tubal ostium) marks the entrance to the oviduct. This papilla was also scuffed and hemorrhagic in the hour after mating, but had regenerated its epithelium 24 hours

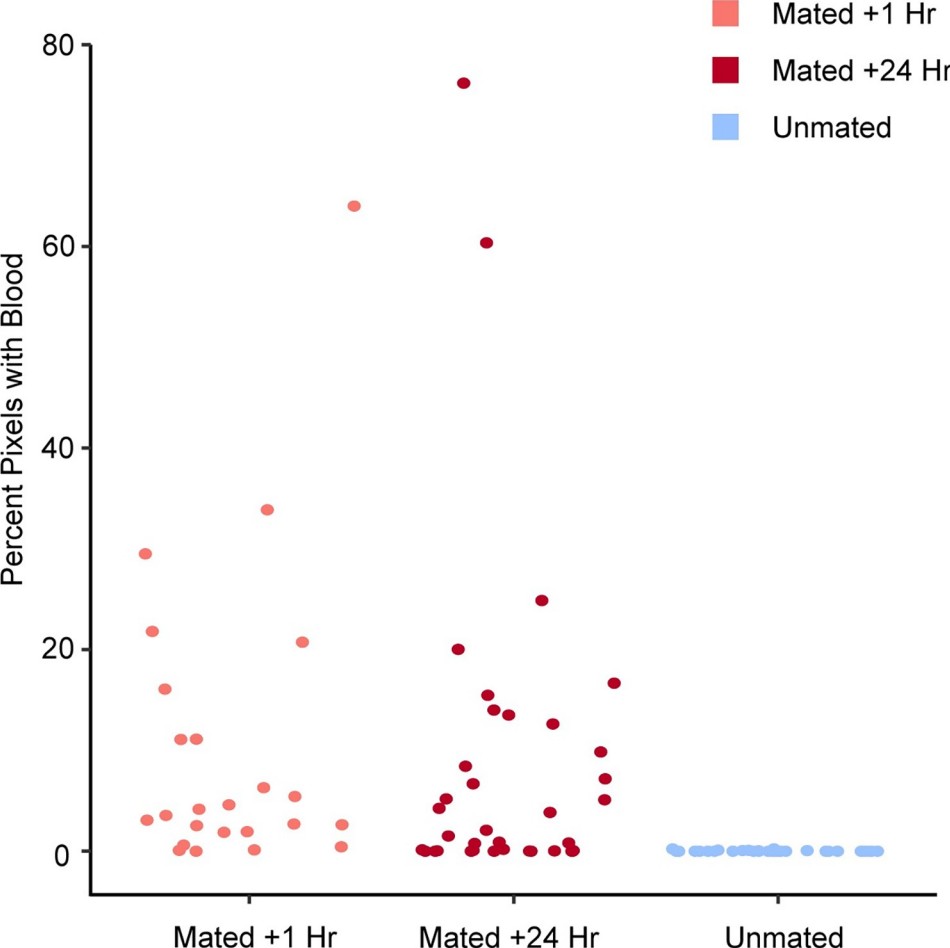

**Fig 3. Total percentages of red pixels indicative of blood.** Unmated alpacas had very minimal blood in their reproductive tract compared to females mated within 1 h and 24 hrs prior to euthanasia.

later (Fig 6). Thus copulatory damage to the reproductive tract was widespread and extended from the cervix to the very tips of both uterine horns.

Neutrophil infiltration (Fig 7) was patchy along the epithelium in a single tissue section. Averaging at least two semi-serial tissue sections of each alpaca's uterine mucosa, neutrophil infiltration was comparable in females from the unmated and mated + 1hr conditions (18.6 +/- 15.3 vs 10.8 +/- 3.1 neutrophils/mm epithelium, respectively). Infiltration appeared somewhat higher in uterine mucosa samples taken 24 hours after mating (40.4 +/- 24.0 neutrophils/mm epithelium). There was notable variability in neutrophil infiltration of the subepithelial zone, not only across multiple slides from the same mucosa but even within a 1.5mm stretch of epithelium in a single section. Consistent with these histological counts of neutrophil infiltration, the density of neutrophils suspended in the lumenal fluid was higher 24 hours after mating than it was in receptive unmated females (Table 1). Variability in neutrophil density was also present in the samples taken from luminal fluid.

## Location of the ejaculate

Immediately after mating sperm was detected throughout the female reproductive tract as expected, though generally there were relatively fewer sperm in the vagina than in the upper

## receptive unmated    mated + 1 hr    mated + 24 hrs

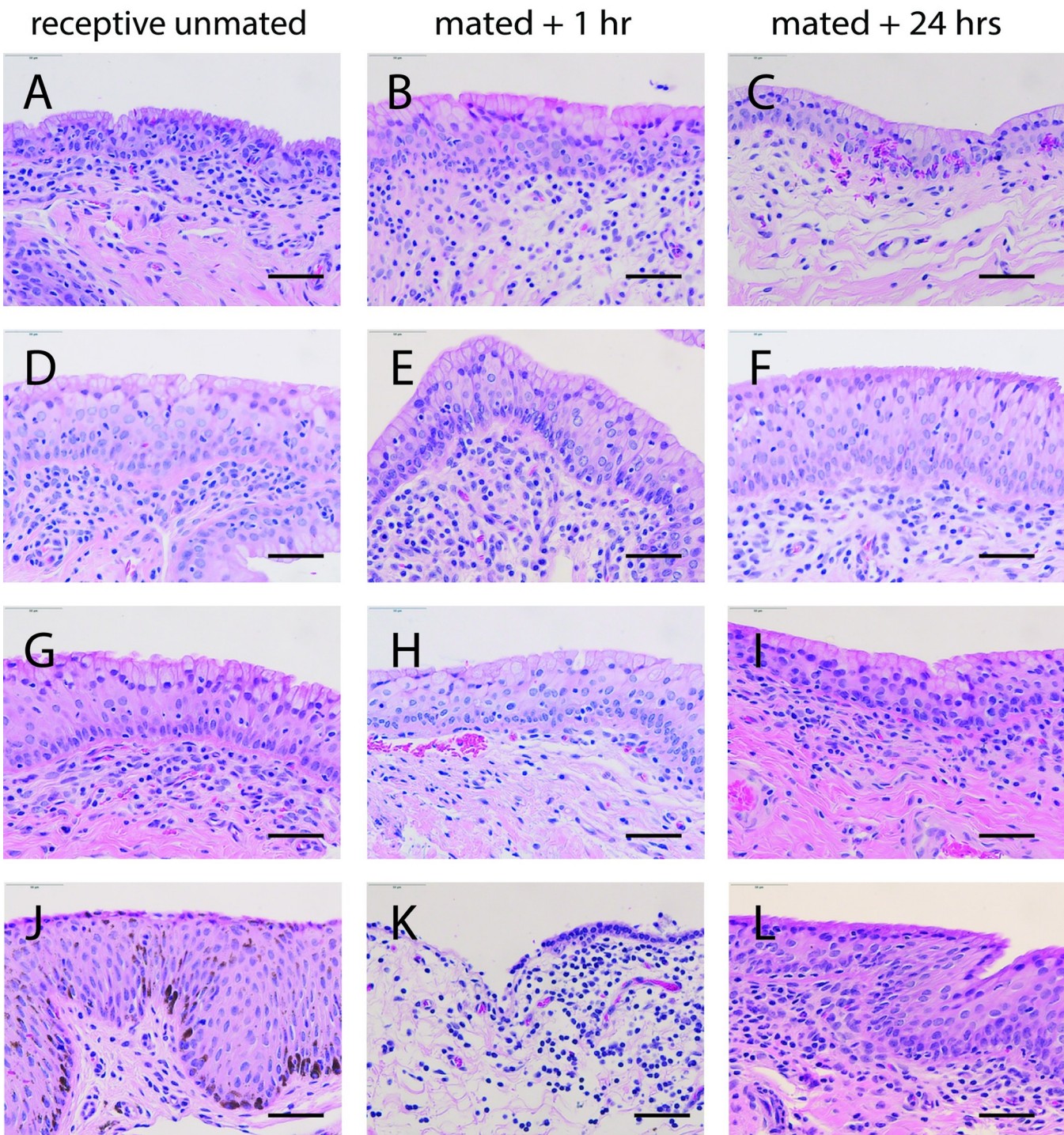

**Fig 4. Epithelium of cranial vagina is usually intact after mating.** Lining of the cranial vagina before and after mating in alpaca stained with H&E. First column n = 4 receptive unmated females (A, D, G, J). Center column: n = 4 mated + 1 hour females (B, E, H, K). Third column: n = 4 mated + 24 hr females (C, F, I, L). Scale bar: 50μm.

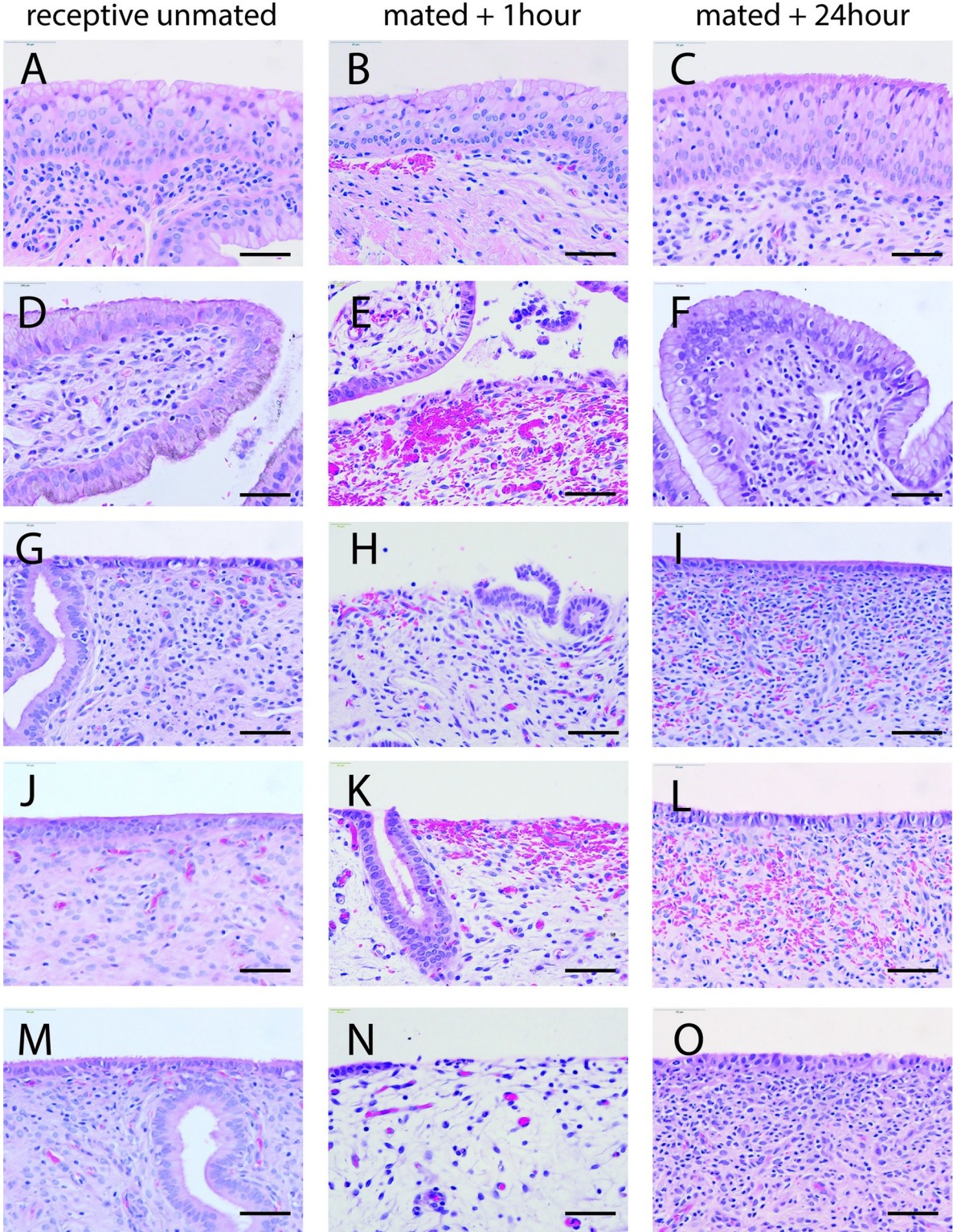

**Fig 5. Reproductive mucosa of alpaca vagina, cervix and uterus before and after mating stained with H&E.** First column: receptive unmated females. Middle column: mated + 1hr. Third column: mated + 24hrs. A, B, C: vaginal epithelium. D, E, F: cervical epithelium. G, H, I: uterine body. J, K, L: right uterine horn. M, N, O: left uterine horn. Scale bar: 50μm.

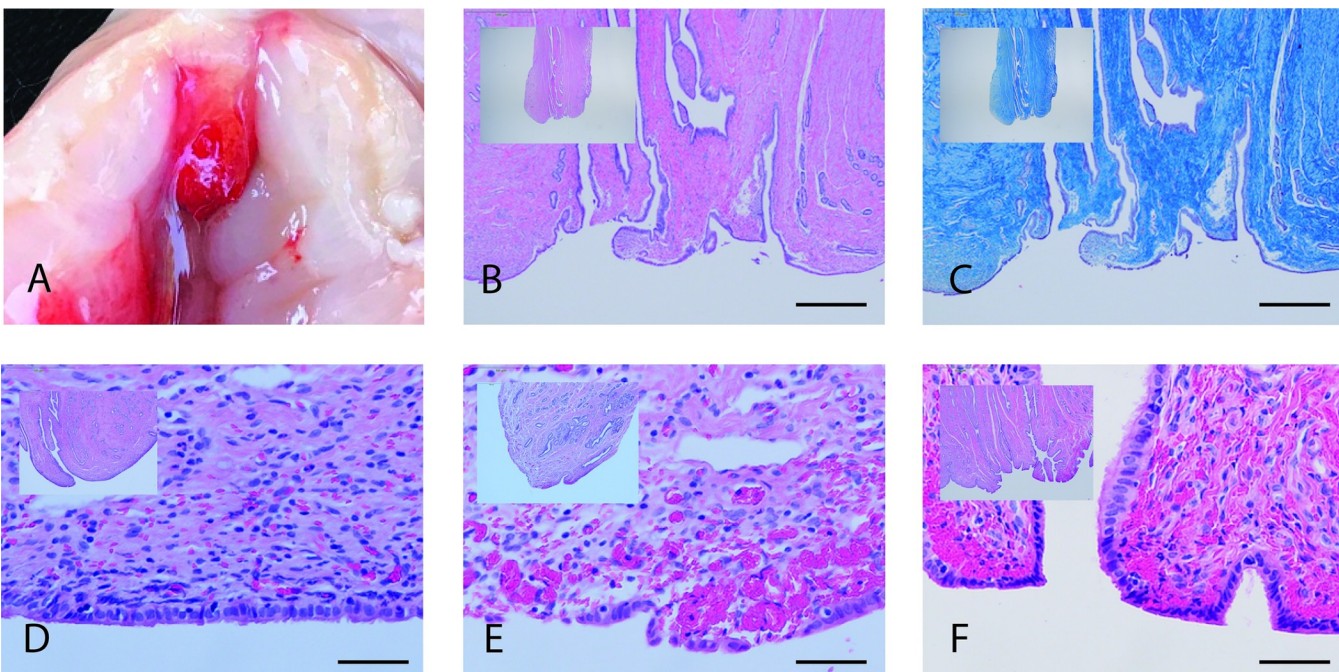

**Fig 6. Uterine tubal ostium (papilla marking entry to oviduct) scuffed and hemorrhagic after mating in alpaca.** Top row: (A) Gross anatomy showing a hemorrhagic oviductal papilla at the tip of one uterine horn within an hour after mating. B & C: histology of the oviductal papilla tip showing the folded, convoluted oviductal lumen by H&E (B) and Masson's trichrome for connective tissue (C). Scale bar (A & B) 500um. D, E, F: oviductal papilla from (D) unmated receptive female, (E) mated + 1 hr female and (F) mated + 24 hr female. Scale bar (D, E, F) 50μm.

regions of the female tract. Sperm was relatively abundant inside the oviducts of some mated females within one hour after mating, suggesting some semen is ejaculated directly in the uterine papillae of the uterotubal junction and from there it quickly enters the oviducal lumen (Table 2). Most females examined 24 hrs after mating had no sperm in their uterus, cervix or vagina, with sperm remaining only in the oviducts. The exception was one female (F035) that still had some sperm throughout her tract.

## Discussion

Here we report that the alpaca penis travels through the entire female reproductive tract during copulation and deposits semen at the uterine papillae of the oviduct entrance, where sperm

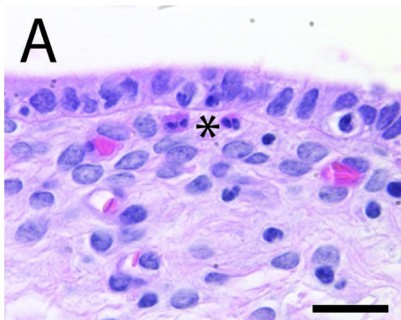
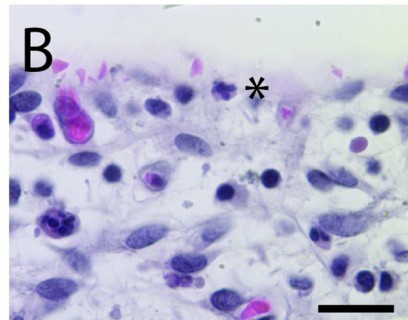
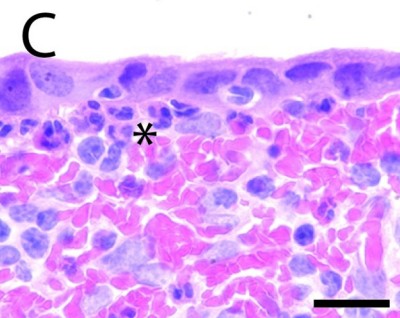

**Fig 7. Neutrophil infiltration of epithelium and subepithelial zone of alpaca uterus.** Neutrophils, identified by their pinkish cytoplasm and lobed nucleus, were scarce in the unmated, receptive (panel A) and mated + 1hr condition (B) but plentiful in the mated + 24 hr condition (C). Sections stained with H&E. Asterix (*) highlights neutrophil in image. Scale bar 25μm.

**Table 1. Qualitative relative abundance estimates of Neutrophil (N) and Leukocytes (L) from smear samples in different regions of the alpaca female reproductive tract.**

| Mating condition | Alpaca ID | vagina | cervix | uterine body | L horn | R horn | L oviduct | R oviduct |
|---|---|---|---|---|---|---|---|---|
| receptive unmated | A040 | • | 1N/0L | 1N/1L | 1N/1L | 1N/1L | 1N/0L | 1N/0L |
| receptive unmated | A041 | 1N/1L | 1N/1L | 0N/0L | • | 0N/1L | 0N/0L | 0N/0L |
| receptive unmated | A043 | 4N/2L | 4N/0L | 0N/0L | 1N/0L | 1N/1L | 0N/0L | 0N/0L |
| receptive unmated | A044 | 1N/0L | 0N/0L | 1N/0L | 1N/0L | 0N/0L | 1N/0L | 1N/0L |
| mated + 24hr | A030 | 1N/1L | 1N/1L | 2N/2L | 0N/1L | 1N/1L | 0N/3L | 0N/3L |
| mated + 24hr | A033 | 2N/0L | 2N/2L | 2N/2L | 0N/1L | 1N/3L | 0N/2L | 1N/1L |
| mated + 24hr | A035 | 1N/0L | 2N/2L | 1N/2L | 1N/1L | 1N/2L | 1N/1L | 1N/1L |
| mated + 24hr | A039 | 3N/2L | 4N/2L | 4N/1L | 4N/2L | 1N/2L | • | 0N/0L |
| mated + 24hr | A042 | 1N/1L | 4N/1L | 3N/1L | 1N/1L | 1N/1L | 1N/1L | 1N/1L |

Scale of 0–4, where 0 = no detectable immune cells, 1 = 1–25% of the slide was covered with immune cells, 2 = 25–50%, 3 = 50–75% and 4 = 75–100%, and • indicates missing data. Immune cells appear relatively more abundant in females mated 24 hr prior, compared to unmated females, but individual variation is high.

quickly enter the oviducts. Although our sample size is relatively small, there was gross and histological evidence of hematoma (hemorrhage) in the reproductive mucosa from the cervix all the way to the very tips of the uterine horns at the opening to the oviduct, while no such damage was evident in any of the unmated females. Sperm were also identified in the oviducts within an hour after mating. While intrauterine insemination is often reported as factual in the existing alpaca literature, there has been no conclusive evidence for it. Our report here (in combination with Brennan [12]), demonstrates histologically and functionally that intromission in the alpaca is much deeper than the uterus, reaching the uterine papillae. Copulatory wounding of the mucosa extends from the cervix to the very tips of the uterine horns, while largely sparing the vagina. We have extended the speculated depth of penetration to the very opening of the oviduct at the tips of the horns and sampled in the immediate post-copulatory period to establish that sperm are present in the oviduct very quickly after mating. Intrauterine penile intromission is extremely rare in mammals, and the extreme depth of penetration we observed may have some interesting evolutionary consequences in alpaca.

**Table 2. Qualitative estimates of relative sperm abundance from smear samples in different regions of the female reproductive tract of females mated 1 and 24 hours prior to examination.**

| mating condition | alpaca ID | copulation duration, min | sperm abundance | | | |
|---|---|---|---|---|---|---|
| | | | vagina | cervix | uterus L/R | oviduct L/R |
| mated + 1hr | A031 | 28 | 1 | 3 | 1/1 | 4/4 |
| mated + 1hr | A032 | 12 | 1 | 3 | 2/1 | 2/0 |
| mated + 1hr | A024 | 13 | 1 | 0 | 1/3 | 4/4 |
| mated + 1hr | A025 | 20 | 3 | 1 | 2/2 | 1/0 |
| mated + 24hr | A030 | unknown | 0 | • | 1/0 | 1/0 |
| mated + 24hr | A033 | 15 | 0 | 0 | 0/0 | 0/0 |
| mated + 24hr | A035 | 20 | 1 | 2 | 0/0 | 3/1 |
| mated + 24hr | A036 | 30 | 0 | 0 | 0/0 | 0/1 |
| mated + 24hr | A039 | 15 | 0 | 0 | 0/0 | •/0 |
| mated + 24hr | A042 | 25 | 0 | 1 | 0/0 | 1/1 |

Scale of 0–4, where 1 = 1–25% of the slide was covered with sperm/immune cells, 2 = 25–50%, 3 = 50–75% and 4 = 75–100%. Sperm were found in the upper regions of the female reproductive tract, including the oviducts in all females with the exception of the majority of the +24 hour group. • = no specimen.

Within an hour after mating, widespread bleeding was present in the upper reproductive tract, but was minimal in the vagina. The cylindrical alpaca vagina is much wider than the penis, so the interaction between the penis and the vagina is likely minimal [12] resulting in little abrasion during copulation. This suggests that for most of the average 20 minute copulation, the tip of the penis is likely in one or the other uterine horn rather than in the vagina. One exception to this pattern was female 36, where blood was only observed in the hymen and vagina, and none in the cervix or uterus/uterine horns. While this female mated for 30 minutes, it is possible that the penis was unable to pass through the cervix during this time, an observation that is supported by the small quantity of sperm found in her oviducts the day after mating. In general, both the abrasion and the discovery of semen at the tip of both uterine horns indicates that the alpaca penis is repositioned during copulation, to move from one horn to the other. Previously we described that the lumen of the cervix and uterine horns have multiple constrictions as evidenced by 3-D models made on intact females [12]. In the present study, we see mating-induced hematoma and abrasion at each of these points of constriction in the uterine horns.

Our histological results show that mating abraded the mucosa, stripping swathes of luminal epithelium from the cervix and uterine horns, all the way up the tract to the uterine papillae marking the entrance to the oviducts. Blood had leaked into the underlying mucosa. Within 24 hours a simple cuboidal epithelium re-lined the uterine body, horns, and oviductal papilla. Red blood cells were often still visible in the underlying stromal tissue, suggesting that while the wounded epithelial tissue is rapidly regenerated, phagocytosis and clearance of red blood cells in the deeper tissue is a lengthier process in the first 24 hours after mating. Neutrophils had flocked to the subepithelial zone in the 24 hours following mating, consistent with their early role in inflammation and wound healing, and were also found in the lumenal fluid after migrating across the tract epithelium. Hematoxylin and eosin staining made the identification of other leukocyte subtypes (lymphocytes, macrophages) difficult.

How might a system of intrauterine intromission and copulatory wounding have evolved? There are multiple possibilities that include facilitating the introduction of ovulation inducing factors, transient inflammation to support early pregnancy, and/or sexual conflict over copulation. Alpacas, like other camelids such as the llama (*Lama glama*), Bactrian camel (*Camelus bactrianus*) and dromedary camel (*Camelus dromedarius*) are induced ovulators. In receptive female alpacas and llamas with mature follicles, serum luteinizing hormone begins to rise during copulation and peaks 3–4 hours later [16]. Typically, induced (or "reflex") ovulation is linked to the physical stimulation of mating, where sensory afferents trigger the neuroendocrine release of LH [17]. In camelids, however, ovulation is induced by a chemical factor in semen (Bactrian camel: [18; 19]; llamas and alpacas: [20]). The ovulation inducing factor in camelid seminal plasma is ß Nerve Growth Factor (ß-NGF) [21], the most abundant protein in alpaca seminal plasma [22]. In alpacas, intramuscular injection of ß-NGF induces ovulation [22] and has a luteotrophic effect similar to that of the GnRH analogue Buserelin [22, 23]. During intrauterine intromission, the alpaca penis (either its sharp urethral process or its cartilaginous sickle-shaped tip) mechanically abrades the epithelium and the underlying stromal tissue, breaking blood vessels in the process. One effect of the abrasion could be to provide ß-NGF greater access to the female's circulatory system. Indeed Ratto et al. [24] showed that the ovulation rate following intrauterine infusion of seminal fluid was increased if the endometrial mucosa had been lightly scraped by the tip of a plastic pipette ahead of time. Likewise Tibary and Annouassi [19] report that in the bactrian camel, intrauterine deposit of semen was 87% effective in inducing ovulation, while natural mating was 100% effective [19]. Adams et al. [20] similarly noted that while intramuscular injection of seminal plasma induced ovulation in 13/14 alpacas (93%), intrauterine administration alone did not (0/12, 0%), leading them to suggest

that "perhaps absorption of OIF in seminal plasma subsequent to natural mating is facilitated by the hyperemia of the excoriated endometrium." Thus perhaps it is most precise to describe a physiochemical basis for ovulation in camelids; a crucial ovulation-inducing chemical factor in semen whose action is augmented by copulatory wounding of the reproductive mucosa.

Because inflammation in the reproductive tract is a known risk factor for miscarriage and preterm birth [25], we might consider it to be a steep evolutionary cost of copulatory wounding. A large body of evidence, however, demonstrates that short-term immune activation promotes early pregnancy. Choreographed inflammation of the uterine mucosa is a universal response to copulation in mammals [26] including marsupials [27]. Endometrial wounding has even been explored clinically as a treatment for cycle failure in IVF [28]. In several studies, superficial endometrial scratching with a biopsy catheter improved the rate of successful embryo transfer in humans [29–32], though evidence of its efficacy in cases of infertility is not without controversy [33].

The normal post-copulatory inflammation that follows mating in mammals shares key features with the classic inflammatory cascade that results from wounding or microbial infection [34]. In the first hours after ejaculation an acute influx of leukocytes into the subepithelial stroma of the uterus eliminates dead and dying sperm along with any pathogens carried within the seminal fluid [26]. Most of the early responding leukocytes are neutrophils. In alpaca, natural mating led to neutrophil infiltration of the subepithelial zone as well as the extrusion of neutrophils into the uterine lumen. In humans, as the transient inflammation resolves, immunomodulatory macrophages persist in the uterus [35–37]. Macrophages can work in several ways to support tissue repair and angiogenesis [38]. They could also be involved in inducing immunological tolerance to the fetus.

Post-copulatory mucosal inflammation resolves quickly, and the capacity for very rapid resolution was likely key to the evolution of extended pregnancy in eutherians [27]. Despite its transience, the inflammation imprints longer-lasting functional changes on uterine immune cells whose activity supports the events of early pregnancy [39, 40]. For example in mice, post-copulatory inflammation induced by exposure to seminal fluid is followed by the expansion of T regulatory cells that induce immunological tolerance to the fetus [41, 42]. Tolerance toward paternal alloantigens may be especially important in animals like humans and rodents where very invasive placentation means that fetally-derived trophoblast cells are exposed to maternal immune effectors within the uterine stroma. In alpaca, however, the trophoblast is noninvasive. The trophoblast of the chorion remains intact, interdigitating with the intact uterine epithelium [43, 44]. While the presence and activity of T cells in camelids has not been studied to date, the lack of trophoblast invasion in alpaca may blunt the need for immune tolerance of paternal alloantigens in the uterus.

Signaling molecules associated with macrophages are also important in early uterine receptivity and in embryo development. In humans and mice, macrophages produce the cytokines IL-1B and LIF that enhance uterine receptivity to the blastocyst [45, 46]. The macrophage-associated cytokine GM-CSF, a key element of the inflammatory cascade, promotes glucose transport in mouse [47] and human blastomeres [48]. Macrophages can also support tissue remodeling and angiogenesis [49] through their production of matrix metalloproteinases, or MMPs [50]. MMPs are important in the uterine modifications that support angiogenesis and placentation. In alpacas MMPs are differentially expressed in the left and right uterine horns [51], which is interesting because while the chorion interdigitates with the epithelium in both horns, the embryo is carried on the left approximately 90% of the time.

Mammals use a range of copulatory signals to induce transient inflammation in the uterus, suggesting that the co-option of inflammation may have been important enough to reproduction that multiple mechanisms evolved to induce it. Mice are perhaps best understood. In

mice, seminal fluid contains the factor TGF-ß1, which induces the influx of inflammatory cells into the uterus [52]. TGF-ß1 stimulates the production of the pro-inflammatory signaling molecule GM-CSF [52] by uterine epithelial cells, which in turn targets stromal macrophages, granulocytes and dendritic cells [53]. But mammals can use a range of mating-associated stimuli to reach the same inflammatory outcome, as demonstrated in domestic ungulates. In cattle, sperm aggregate in intrauterine glands after mating and induce the production of the inflammatory cytokines IL8, TNF alpha, and IL1ß by the glandular epithelium [54, 55]. In pigs, seminal fluid is sufficient to induce neutrophil infiltration of the estrogen-primed endometrium [56], as well as recruitment of macrophages and dendritic cells [57]. Activated T cells accumulate as a result of the pig conceptus' production of the inflammatory molecule IFN-gamma [58]. Together, these induce pro-inflammatory changes in the uterus. Such signals from alpaca sperm or seminal fluid have yet to be elucidated, and since the alpaca uterine lining is quite literally excoriated by penile intromission, wounding itself may be sufficient to induce an inflammatory response in the underlying tissue. Potential longer-term effects of this early inflammation—effects on macrophage or T cell function that could influence uterine receptivity, embryo health or placental development—are unknown in alpaca.

In short, the co-option of the inflammatory response was probably instrumental to the evolution of pregnancy in therians, and its quick resolution important to the evolution of extended pregnancy in eutherians [27, 59]. Copulatory wounding, as we see in alpaca, would be a way to induce this short-term inflammatory response that could lead to functional changes in uterine immune cells such as macrophages. If the physiology of early pregnancy in alpaca is similar to what we know of other mammals, the longer-term effect of this wounding-induced inflammation could potentially support endometrial receptivity, blastocyst development, and the tissue remodeling necessary for placental development.

Copulatory wounding is often associated with sexual conflict, for example in seed beetles (*Callosobruchus sp)* and bed bugs (Cimicidae) [60], where males either scrape the female reproductive lining with spines in their intromittent organ or perforate her exoskeleton with their needle-like intromittent organ to introduce seminal fluid and sperm directly into her body cavity. However, male copulatory wounding can also cause infection (e.g. in Drosophila, [61], result in blood loss, and in the introduction of foreign particles that elicit an immune response [60]. These male adaptations that cause harm to the female are known to be beneficial for the male in that they can reduce the likelihood of female remating or increase her refractory period, and/or increase the direct transfer of seminar proteins to her plasma thereby reducing the possibility that females can exert cryptic female choice [60]. However, it seems unlikely that alpacas experience the usual selective pressures that may induce copulatory wounding as a result of sexual conflict; females typically do not mate with multiple males because a single dominant male controls reproductive access to the female group, and female cooperation and receptivity are required for copulation to succeed because females must adopt a prone position to allow intromission, and the lengthy copulation would allow females plenty of opportunity to reject the male by changing position. Either males or females may terminate the copulation, but usually the male stops anywhere between 5–60 minutes after intromission, by standing up. Other times the female may push up or roll to her side and terminate copulation.

Our results are in accordance with Bravo et al.'s [6] data suggesting that in alpacas the uterotubal junction acts as a sperm reservoir, much the way the cranial vagina and cervix function in organisms with intravaginal intromission. We detected sperm in the oviductal isthmus within an hour of copulation and found subepithelial blood effusion and scuffing of the oviductal papilla. How could the timing of sperm transport relative to ovulation work? Bravo [6] reported that the bulk of sperm in the tract was found in the utero-tubal junction between 6

and 12 hours post-copulation, moving up into the oviductal isthmus by 18 hours post-copulation. Alpacas ovulate approximately 26–30 hours after either copulation [62] or the intramuscular injection of ß-NGF [23], so sperm would be present in the isthmus at the time of ovulation and fertilization should occur quickly thereafter.

Alpaca sperm collection is famously difficult, with small volumes (< 2mL) and high percentages of nonviable sperm typically reported, particularly in the first few minutes of copulation (13.8 vs. 37.1% live sperm at early vs. later collection time points [10]. In addition, the testes of alpaca are relatively small given its body size in comparison with other livestock [63]. Small ejaculate volume may be an evolutionary consequence of intrauterine insemination in that sperm are not lost through the reproductive tract as it is the case in most mammals, but rather deposited right next to the oviduct. Therefore selection for high volume, motility, density and viability of sperm may be reduced in alpacas. Sperm in the cervix and uterus may be carried by the penis when the male switches the location of intromission from one horn to the other, or it may be the result of the "dribble ejaculation" pattern reported in alpaca during semen collection [9]. We had previously attempted to visualize the uterus with external ultrasound but the pelvic girdle prevents external visualization, so we did not try to use this technique during copulation.

The wounding we document, showing interruption of the epithelial barrier and blood effusion in the underlying mucosa, may ensure timely ovulation by delivering the seminal ovulation-inducing factor NGFß into the female alpaca's blood stream. Rather than being induced by purely chemical, or purely physical stimuli, in alpacas it may be a combination: physical wounding to augment the delivery of an ovulation-inducing factor in seminal plasma. Thus a peculiar copulatory mode in alpaca may improve the odds of successful fertilization and pregnancy. The fact that the entire reproductive tract in alpaca is essentially functioning as a vagina during copulation likely has some immunological consequences worth further investigation.

## Supporting information

**S1 Table. Blood per region.** Percentage of pixels with blood in each region of the reproductive tract of unmated females, and females mated 1 hr or 24 hrs before photographing the tract.
(PDF)

**S2 Table. Neutrophil mucosal counts.**
(XLSX)

**S3 Table. Tract cytology and sperm counts.**
(PDF)

**S1 Fig. Photos of mated and unmated alpacas used for Image J analysis.**
(PDF)

## Acknowledgments

We are grateful to Morning Beckons Farm for allowing access to their animals and for kindly providing specimens for this research. Dr. Rachel Keeffe helped make Fig 3, and Dr. Brandon Hedrick helped with statistical analysis and Fig 2. We are also grateful to Prof. Diana Perez-Staples, and Prof. Martha Valdivia for their helpful reviews of an earlier version of this manuscript.

## Author Contributions

**Conceptualization:** Patricia L. R. Brennan.

**Data curation:** Patricia L. R. Brennan, Stephen Purdy, Sarah J. Bacon.

**Formal analysis:** Patricia L. R. Brennan, Stephen Purdy, Sarah J. Bacon.

**Funding acquisition:** Patricia L. R. Brennan, Stephen Purdy.

**Investigation:** Patricia L. R. Brennan, Stephen Purdy, Sarah J. Bacon.

**Methodology:** Patricia L. R. Brennan, Stephen Purdy, Sarah J. Bacon.

**Project administration:** Patricia L. R. Brennan.

**Resources:** Patricia L. R. Brennan.

**Validation:** Sarah J. Bacon.

**Visualization:** Sarah J. Bacon.

**Writing – original draft:** Patricia L. R. Brennan, Sarah J. Bacon.

**Writing – review & editing:** Patricia L. R. Brennan, Stephen Purdy, Sarah J. Bacon.

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
