## [Decision Letter · Decision Letter 0]

18 Jan 2024

PONE-D-23-40086Intra-horn insemination in the alpaca Vicugna pacos:  Copulatory wounding and deep sperm depositionPLOS ONE

Dear Dr. Bacon,

Thank you for submitting your manuscript to PLOS ONE. After careful consideration, we feel that it has merit but does not fully meet PLOS ONE’s publication criteria as it currently stands. Therefore, we invite you to submit a revised version of the manuscript that addresses the points raised during the review process.

We look forward to receiving your revised manuscript.

Kind regards,

Jayonta Bhattacharjee

Academic Editor

PLOS ONE

Journal Requirements:

4. Thank you for stating the following financial disclosure: "Mount Holyoke College faculty research grant to PLRB.

NSF CAREER grant to PLRB, IOS:2042260"

Reviewers' comments:

Reviewer's Responses to Questions

**Comments to the Author**

1. Is the manuscript technically sound, and do the data support the conclusions?

Reviewer #1: Yes

Reviewer #2: Yes

2. Has the statistical analysis been performed appropriately and rigorously? 

Reviewer #1: Yes

Reviewer #2: Yes

3. Have the authors made all data underlying the findings in their manuscript fully available?

Reviewer #1: Yes

Reviewer #2: Yes

4. Is the manuscript presented in an intelligible fashion and written in standard English?

Reviewer #1: Yes

Reviewer #2: Yes

5. Review Comments to the Author

Reviewer #1: Review of Intra-horn insemination in the alpaca Vicugna pacos: Copulatory wounding and deep sperm deposition. This is an interesting manuscript that has novel data on the process of copulation and sperm transfer in the alpaca. The manuscript is mostly descriptive, but it is very detailed and interesting to read. The results have implications for the evolution of intromission behaviour, the evolution of female and male gonads and sexual conflict theory. I have only some minor comments.

Given the wide audience of Plos one, the introduction would greatly benefit from a diagram of the alpaca reproductive system.

Materials and methods

Local farm where?

Examined and photographed immediately after euthanized?

It may help other researchers to know what problems were faced with sample preservation of slides, did the stain not work? That is important information so other researchers don’t make the same mistake.

Three values after the 0 for p values is more than enough.

I suggest values for Table 1 should be a graph, with averages or percentages, making data easier to visualize.

Discussion, “female cooperation and receptivity are required for copulation to succeed because copulation is prolonged”not quite following the argument as to why female cooperation is necessary because copulation is long.. In insects copulations can be very lengthy, much more than the 20 min listed here for alpacas, but there still can be sexual conflict if the male is coercing or manipulating the female into mating with him… So I would suggest perhaps limiting this discussion to the ways in which females need to adapt certain positions to allow intromission.

Also, is there any evidence for females controlling copula duration as in tephritid flies?

Reviewer #2: This is an interesting study of the alpaca natural insemination, there are some mistakes and clarifications and I included my comments in the article PDF attached with this format, for this publication, it is important to know how the animals were cared for this study, the ethic component.

6. PLOS authors have the option to publish the peer review history of their article (what does this mean?). If published, this will include your full peer review and any attached files.

Reviewer #1: **Yes: **Diana Perez-Staples

Reviewer #2: **Yes: **Martha Valdivia

---

## [Author Response · Author response to Decision Letter 0]

23 Feb 2024

Response to reviewers

Brennan, Purdy and Bacon

Re: Resubmission of MS PONE-D-23-40086

 The reviewers had very minimal comments and we have addressed all of them below. 

R1: Given the wide audience of Plos one, the introduction would greatly benefit from a diagram of the alpaca reproductive system.

AU: We already have a labeled photo in Fig 1, therefore we did not add a diagram.

R1: Local farm where?

AU: Connecticut. Added

R1: Examined and photographed immediately after euthanized?

AU: Yes, added

R1: It may help other researchers to know what problems were faced with sample preservation of slides, did the stain not work? That is important information so other researchers don’t make the same mistake.

AU: We realized that these slides were missing because on the females mated 1hr before euthanasia we had not started yet to use the diff quick stain that allowed us to see immune cells but were using eosin nigrosin stain for live/dead sperm. We have deleted the sentence and instead rewrote this paragraph in the manuscript as follows: “We then used a glass slide to touch each region of the inside of the reproductive tract and stained these slides either with a modified Giemsa stain (Differential Quik Stain Kit, VWR, Radnor PA) to look for the presence of inflammatory cells (unmated and mated +24 hrs) and sperm (mated +24 hrs), or with Semen Morphology Stain (eosin/nigrosin) (mated +1hr)”.

R1: Three values after the 0 for p values is more than enough.

AU: Fixed

R1: I suggest values for Table 1 should be a graph, with averages or percentages, making data easier to visualize.

AU: Added Figure 2. Changed the figure captions for all remaining figures

R1: Discussion, “female cooperation and receptivity are required for copulation to succeed because copulation is prolonged”not quite following the argument as to why female cooperation is necessary because copulation is long.. In insects copulations can be very lengthy, much more than the 20 min listed here for alpacas, but there still can be sexual conflict if the male is coercing or manipulating the female into mating with him… So I would suggest perhaps limiting this discussion to the ways in which females need to adapt certain positions to allow intromission. Also, is there any evidence for females controlling copula duration as in tephritid flies?

AU: We modified this paragraph, now it reads as follows: “However, it seems unlikely that alpacas experience the usual selective pressures that may induce copulatory wounding as a result of sexual conflict; females typically do not mate with multiple males because a single dominant male controls reproductive access to the female group, and female cooperation and receptivity are required for copulation to succeed because females must adopt a prone position to allow intromission, and the lengthy copulation would allow females plenty of opportunity to reject the male by changing position. Either males or females may terminate the copulation, but usually the male stops anywhere between 5-60 minutes after intromission, by standing up. Other times the female may push up or roll to her side and terminate copulation”.

Reviewer #2: This is an interesting study of the alpaca natural insemination, there are some mistakes and clarifications and I included my comments in the article PDF attached with this format, for this publication, it is important to know how the animals were cared for this study, the ethic component.

Reviewer 2 had 8 spelling and formatting corrections and we have fixed all of them. Since R1 also had a question about housing of the alpaca we added the following statement: “We obtained whole reproductive tracts from adult female alpacas that were being culled for meat processing at a farm in Connecticut, USA, where animals are maintained in accordance with State and federal guidelines concerning livestock management in small farms”

---

## [Decision Letter · Decision Letter 1]

17 Mar 2024

PONE-D-23-40086R1Intra-horn insemination in the alpaca Vicugna pacos:  Copulatory wounding and deep sperm depositionPLOS ONE

Dear Dr. Brennan,

Thank you for submitting your manuscript to PLOS ONE. After careful consideration, we feel that it has merit but does not fully meet PLOS ONE’s publication criteria as it currently stands. Therefore, we invite you to submit a revised version of the manuscript that addresses the points raised during the review process.

We look forward to receiving your revised manuscript.

Kind regards,

Jayonta Bhattacharjee

Academic Editor

PLOS ONE

Journal Requirements:

Additional Editor Comments:

Thank you very much for revising the article. I would like to request the authors to address a few minor things. Such as -

Page 8, Line 158: particular (F042, 29 and 23% respectively).

Where are those data, and what is F042. It would be helpful for the readers if they could track and find that information somewhere in the manuscript or in the additional information files.

On the figures, such as in Figures 1 and 2, state that mated, mated day, and unmated. Figure 3 states, mated, mated 24hrs, and unmated. Figure 4 and 5: receptive unmated, mated+1hour, mated+24hour. It would be better to make it uniform style.

Reviewers' comments:

Reviewer's Responses to Questions

**Comments to the Author**

1. If the authors have adequately addressed your comments raised in a previous round of review and you feel that this manuscript is now acceptable for publication, you may indicate that here to bypass the “Comments to the Author” section, enter your conflict of interest statement in the “Confidential to Editor” section, and submit your "Accept" recommendation.

Reviewer #1: All comments have been addressed

Reviewer #2: All comments have been addressed

2. Is the manuscript technically sound, and do the data support the conclusions?

Reviewer #1: Yes

Reviewer #2: Yes

3. Has the statistical analysis been performed appropriately and rigorously? 

Reviewer #1: Yes

Reviewer #2: Yes

4. Have the authors made all data underlying the findings in their manuscript fully available?

Reviewer #1: Yes

Reviewer #2: Yes

5. Is the manuscript presented in an intelligible fashion and written in standard English?

Reviewer #1: Yes

Reviewer #2: Yes

6. Review Comments to the Author

Reviewer #1: The authors have adequately replied to the queries. I have no further observations, we need more studies like this one on the female reproductive tract.

Reviewer #2: Great and interesting research, your publication has improved in good condition and level. Your results showed the important of type of intercourse in alpacas, the low number of sperm in female tract after 24 hours is unbelievable Congrats!!!

7. PLOS authors have the option to publish the peer review history of their article (what does this mean?). If published, this will include your full peer review and any attached files.

Reviewer #1: **Yes: **Diana Pérez-Staples

Reviewer #2: **Yes: **Martha Valdivia

---

## [Author Response · Author response to Decision Letter 1]

19 Mar 2024

The editor requested we change the captions in our figures so they would match and we made the changes.

---

## [Editor Report · Decision Letter 2]

27 Mar 2024

PONE-D-23-40086R2Intra-horn insemination in the alpaca Vicugna pacos:  Copulatory wounding and deep sperm depositionPLOS ONE

Dear Dr. Brennan,

Thank you for submitting your manuscript to PLOS ONE. After careful consideration, we feel that it has merit but does not fully meet PLOS ONE’s publication criteria as it currently stands. Therefore, we invite you to submit a revised version of the manuscript that addresses the points raised during the review process.

We look forward to receiving your revised manuscript.

Kind regards,

Jayonta Bhattacharjee

Academic Editor

PLOS ONE

Journal Requirements:

Additional Editor Comments:

Thank you very much for the revised version.

I do not know if I am making any mistake, still, I feel a discrepancy between the text and the supplied supplementary table 1.

In lines: 152-154:

Author mentioned

“percentage of bloody pixels (up to 24% for both), but the cervix and uterus also had high percentages in one female in particular (Female 042:, 29 and 23% respectively, data in S1 Table)”.

When we go to the S1 table, we see the last row of the supplementary table shows,

“mated +24hrs A042 0.06 0 76.19 60.35 13.51 15.47”

Is it the same data the author meant, then where are those percentages. May be uterus (left and right horn) is 29% (13.51+15.47=28.98), and the cervix is 76%. Is it true?

I am sorry for the misunderstanding. It will be better for the reader to see an easy-understanding table. If there is any discrepancy, I would request the author to put the correct information.

---

## [Author Response · Author response to Decision Letter 2]

28 Mar 2024

Comments to the editor

We apologize for the confusion. The data in the table are correct, and we revised the text of the paper to increase clarity of our findings, and rewrote that paragraph in the results to read as follows:

While there was a lot of variation among females in the percent of bloody pixels present per region (Fig 2), the left and/or right uterine horns had the highest percentage of bloody pixels in 6 of 10 females, while the cervix and uterus had the highest percentages in 3 of 10 females (data in S1 Table). The vagina had the lowest evidence of blood in the reproductive tract in both mated categories but one mated female (F036), had blood only in the hymen and vagina and none in the cervix, uterus or uterine horns (Fig 2, all data in S1 Table, and photos in S1 Fig 1).

We also added the following sentences to the discussion:

This suggests that for most of the average 20 minute copulation, the tip of the penis is likely in one or the other uterine horn rather than in the vagina. One exception to this pattern was female 36, where blood was only observed in the hymen and vagina, and none in the cervix or uterus/uterine horns. While this female mated for 30 minutes, it is possible that the penis was unable to pass through the cervix during this time, an observation that is supported by the small quantity of sperm found in her oviducts the day after mating.

---

## [Editor Report · Decision Letter 3]

1 Apr 2024

Intra-horn insemination in the alpaca Vicugna pacos:  Copulatory wounding and deep sperm deposition

PONE-D-23-40086R3

Dear Dr. Brennan,

We’re pleased to inform you that your manuscript has been judged scientifically suitable for publication and will be formally accepted for publication once it meets all outstanding technical requirements.

Kind regards,

Jayonta Bhattacharjee

Academic Editor

PLOS ONE
---

## [Editor Report · Acceptance letter]

5 Apr 2024

PONE-D-23-40086R3 

PLOS ONE

Dear Dr. Brennan, 

I'm pleased to inform you that your manuscript has been deemed suitable for publication in PLOS ONE. Congratulations! Your manuscript is now being handed over to our production team.

Kind regards, 

on behalf of

Dr. Jayonta Bhattacharjee 

Academic Editor

PLOS ONE